# Genetic features of antimicrobial drug-susceptible extraintestinal pathogenic *Escherichia coli* pandemic sequence type 95

Yuan Hu Allegretti,[1] Reina Yamaji,[2] Sheila Adams-Sapper,[3] Lee W. Riley[1]

**ABSTRACT** Extraintestinal pathogenic *Escherichia coli* (ExPEC) belonging to multilocus sequence type 95 (ST95) is one of the most geographically widespread ExPEC lineages causing bloodstream infection (BSI) and urinary tract infection (UTI). In contrast to other widespread ExPEC sequence types, a large proportion of ST95 strains remains susceptible to all antimicrobial agents used to treat BSI or UTI. We aimed to identify the genomic features of ST95 associated with its relatively high drug-susceptible frequency. We analyzed whole genomes of 1,749 ST95 isolates, 80 from patients with BSI or UTI in Northern California and 1,669 isolates from the EnteroBase database. We first compared whole-genome sequences (WGSs) of 887 drug-susceptible strains and 862 strains resistant to one or more drugs (defined genotypically as strains harboring drug-resistance genes annotated in the ResFinder database) to identify genetic features associated with strains devoid of drug resistance genes. We then conducted a pan-genome-wide association study on human clinical isolates of ST95, which included 553 UTI and BSI ST95 isolates. We found 44 accessory genes to be significantly associated with ST95 strains lacking drug resistance genes. Fifteen of these were not found in any of the WGSs of ST131 ExPEC strains, which are frequently multidrug-resistant. These genes were annotated to encode transporter or transfer systems and DNA repair polymerases. A large proportion of ST95 strains may have evolved to adapt to antibiotic-imposed stresses without acquiring drug resistance genes.

**IMPORTANCE** Despite the increasing prevalence of antibiotic-resistant *Escherichia coli* strains that cause urinary tract and bloodstream infections, a major pandemic lineage of extraintestinal pathogenic *E. coli* (ExPEC) ST95 has a comparatively low frequency of drug resistance. We compared the genomes of 1,749 ST95 isolates to identify genetic features that may explain why most strains of ST95 resist becoming drug-resistant. Identification of such genomic features could contribute to the development of novel strategies to prevent the spread of antibiotic-resistant genes and devise new measures to control antibiotic-resistant infections.

**KEYWORDS** *Escherichia coli*, ExPEC, ST95, drug resistance, pan-genome-wide association study

Antimicrobial drug resistance is one of the most serious global public health concerns currently, threatening the effective prevention and treatment of infectious diseases (1, 2). In particular, several drug-resistant Gram-negative bacterial species belonging to Enterobacteriaceae are designated by the Centers for Disease Control and Prevention (CDC) and the World Health Organization (WHO) as "urgent-threat" or "priority 1, critical" pathogens in need of new treatment and research (2, 3). Among the members of Enterobacteriaceae, *Escherichia coli* (*E. coli*) is the most frequent causative agent for

Address correspondence to Yuan Hu Allegretti, yuan_hu@berkeley.edu.

The authors declare no conflict of interest.

See the funding table on p. 14.

common extraintestinal infections such as urinary tract infection (UTI) and bloodstream infection (BSI) (4). *E. coli* strains that cause such infections are referred to as extraintestinal pathogenic *E. coli* (ExPEC) (5). The prevalence and incidence of infections caused by ExPEC strains resistant to a wide spectrum of antimicrobial agents, including extended-spectrum beta-lactamase (ESBL) drugs, fluoroquinolones, trimethoprim–sulfamethoxazole, and aminoglycosides, have been rapidly increasing worldwide (6–12).

Epidemiological studies have shown that more than 50 % of community and healthcare-associated UTI and BSI reported globally are caused by ExPEC strains belonging to a limited set of lineages, as defined by multilocus sequence typing (MLST) (13, 14). These include sequence types ST10, ST69, ST73, ST95, ST127, ST131, and ST393 (15, 16). Except for ST95, large proportions of these lineages are resistant to antimicrobial agents used to treat UTI or BSI. In particular, ST131, the most commonly reported ExPEC lineage worldwide, is frequently multidrug-resistant (17–19).

Although ST95 strains belong to a small group of ExPEC lineages that cause more than 50% of UTI and BSI cases in most regions of the world (10, 15), its drug resistance frequency is characteristically lower, relative to the other common ExPEC multilocus sequence types (14, 20–23). We previously reported that the carriage of a UTI89-like plasmid called pUTI89* and *fimH* type is associated with the pan-susceptible phenotype of ST95 (21, 22). However, the particular genomic features associated with this drug-susceptible phenotype remain unclear. Here, we generated a whole-genome sequence (WGS) data set comprising ST95 isolates collected from North California and EnteroBase database and compared the WGSs of susceptible ST95 strains to those of ST95 strains that carry drug-resistance genes to identify genetic features associated with drug susceptibility. We believe that this analysis may help shed new light on the compensatory fitness advantage of ST95 under antimicrobial drug stress without acquiring drug resistance genes.

## RESULTS

### Frequency and type of drug resistance genes among ST95 strains

We performed WGS analysis of 36 ST95 isolates that were previously collected from UTI patients in Northern California (14) and compared the sequences to those of 44 BSI ST95 isolates from Northern California (21) we previously reported and 1,669 ST95 WGSs deposited in the EnteroBase database (24). We queried the carriage of recognized drug resistance genes against ResFinder and found that 837 (50%) of 1,669 EnteroBase ST95 isolates and 50 (62%) of 80 Northern California isolates did not contain any recognized drug resistance genes. Among the remaining 862 ST95 isolates, 112 recognized drug-resistant genes for 12 different classes of antibiotics were identified (Table S2). Of these, 25 genes for seven different classes of antibiotics were present in 80 clinical ST95 isolates from Northern California (Table 1). These classes of antimicrobial agents included aminoglycosides, beta-lactams, phenicols, trimethoprim, sulfonamide, tetracycline, and macrolides. Genes associated with fosfomycin, lincosamide, colistin, quinolone, and rifampicin resistance were not observed among any of the ST95 isolates from Northern California. In the period when the total annual submission of ST95 isolates to EnteroBase was more than 50 (2010–2017), the prevalence of strains carrying drug-resistant genes has remained relatively constant (33%–59%). When stratified by BSI or UTI, 86 (38%) of 225 BSI isolates and 132 (53%) of 248 UTI isolates were found to be drug-resistant based on the ResFinder database.

Among the sequences from the 80 Northern California isolates, we found extended-spectrum beta-lactamase (ESBL) genes $bla_{CTX-M-14}$, $bla_{CTX-M-15}$, $bla_{TEM-1a}$, $bla_{TEM-1b}$, and $bla_{TEM-1c}$ in one (1.3%), one (1.3%), three (3.9%), 21 (27 %), and one (1.3%) of 80 isolates, respectively. The same set of ESBL genes was present in 29 (1.7%), 44 (2.6%), 18 (1.1%), 335 (20.1%), and 148 (8.9%) of 1,669 isolates among the sequences deposited in the EnteroBase database, respectively. The temporal distribution of drug resistance genes in all the 1,749 ST95 isolates is shown in Table S3.

**TABLE 1** Frequency of antimicrobial resistance genes among *E. coli* isolates belonging to ST95 registered in the EnteroBase database and obtained in Northern California, identified from ResFinder

| Antibiotic class | Resistance gene name | EnteroBase total (%) | North California total (%) |
|---|---|---|---|
| Pan susceptible | | 837 (50) | 50 (62) |
| Aminoglycoside | aac(3)-IId_1 | 36 (2.16) | 2 (2.56) |
| | aadA2_1 | 39 (2.34) | 2 (2.56) |
| | aadA5_1 | 33 (1.98) | 4 (5.13) |
| | ant(2")-Ia_1 | 2 (0.12) | 2 (2.56) |
| | ant(3")-Ia_1 | 38 (2.28) | 1 (1.28) |
| | aph(3")-Ib_5 | 306 (18.33) | 6 (7.69) |
| | aph(6)-Id_1 | 254 (15.22) | 7 (8.97) |
| Beta-lactam | blaCTX-M-14_1 | 29 (1.74) | 1 (1.28) |
| | blaCTX-M-15_1 | 44 (2.64) | 1 (1.28) |
| | blaTEM-1A_1 | 18 (1.08) | 3 (3.85) |
| | blaTEM-1B_1 | 335 (20.07) | 21 (26.92) |
| | blaTEM-1C_1 | 148 (8.87) | 1 (1.28) |
| Phenicol | catA1_1 | 16 (0.96) | 2 (2.56) |
| | floR_2 | 17 (1.02) | 1 (1.28) |
| Trimethoprim | dfrA12_8 | 33 (1.98) | 2 (2.56) |
| | dfrA17_1 | 63 (3.77) | 5 (6.41) |
| | dfrA5_1 | 153 (9.17) | 2 (2.56) |
| Sulfonamide | sul1_2 | 1 (0.06) | 1 (1.28) |
| | sul1_5 | 107 (6.41) | 5 (6.41) |
| | sul2_2 | 278 (16.66) | 4 (5.13) |
| | sul2_3 | 114 (6.83) | 3 (3.85) |
| Tetracycline | tet(A)_6 | 211 (12.64) | 8 (10.26) |
| | tet(B)_1 | 1 (0.06) | 1 (1.28) |
| | tet(B)_2 | 107 (6.41) | 5 (6.41) |
| Macrolide | mph(A)_2 | 42 (2.52) | 3 (3.85) |

## Pan-genome analysis

A total of 35,134 genes constituted the pan-genome of 1,749 ST95 WGSs. Of 35,134 genes, 3,588 (10.2%) were shared among more than 95% of the isolates (core genes) and 31,546 (89.5%) were distributed among subsets of the isolates (accessory genes). Of the latter, 29,522 genes were found in <15% of the isolates (cloud genes). Of the cloud genes, 13,970 (40%) genes were unique to only one isolate.

Among the 862 drug-resistant isolates, 28,388 genes constituted the pan-genome. Of 28,388 genes, 3,616 were core genes and 24,722 were accessory genes. Among the 887 drug-susceptible isolates, 26,425 constituted the pan-genome. Of these, 3,645 were core genes and 22,780 were accessory genes.

## Phylogenetic and principal component analyses

Maximum-likelihood phylogenies of the single-nucleotide polymorphism (SNP) alignment of core genomes and the presence and absence of accessory genomes were obtained from FastTree (25) with the GTR model (Fig. 1). Clustering based on *fimH* types was observed, but we did not observe a strong correlation of drug resistance with phylogenetic trees based on the SNPs of core genes and the presence and absence of accessory genes.

Principal component analysis (PCA) was also conducted to visually assess variables accounting for the variance. Cluster patterns based on the variance of SNPs and accessory genes were associated with the carriage of plasmid pUTI89* and with *fimH* types 18, 27, 30, 41, and 54 in core genomes and accessory genomes (Fig. 2). However, the cluster patterns were not observed with drug resistance or pan-susceptibility.

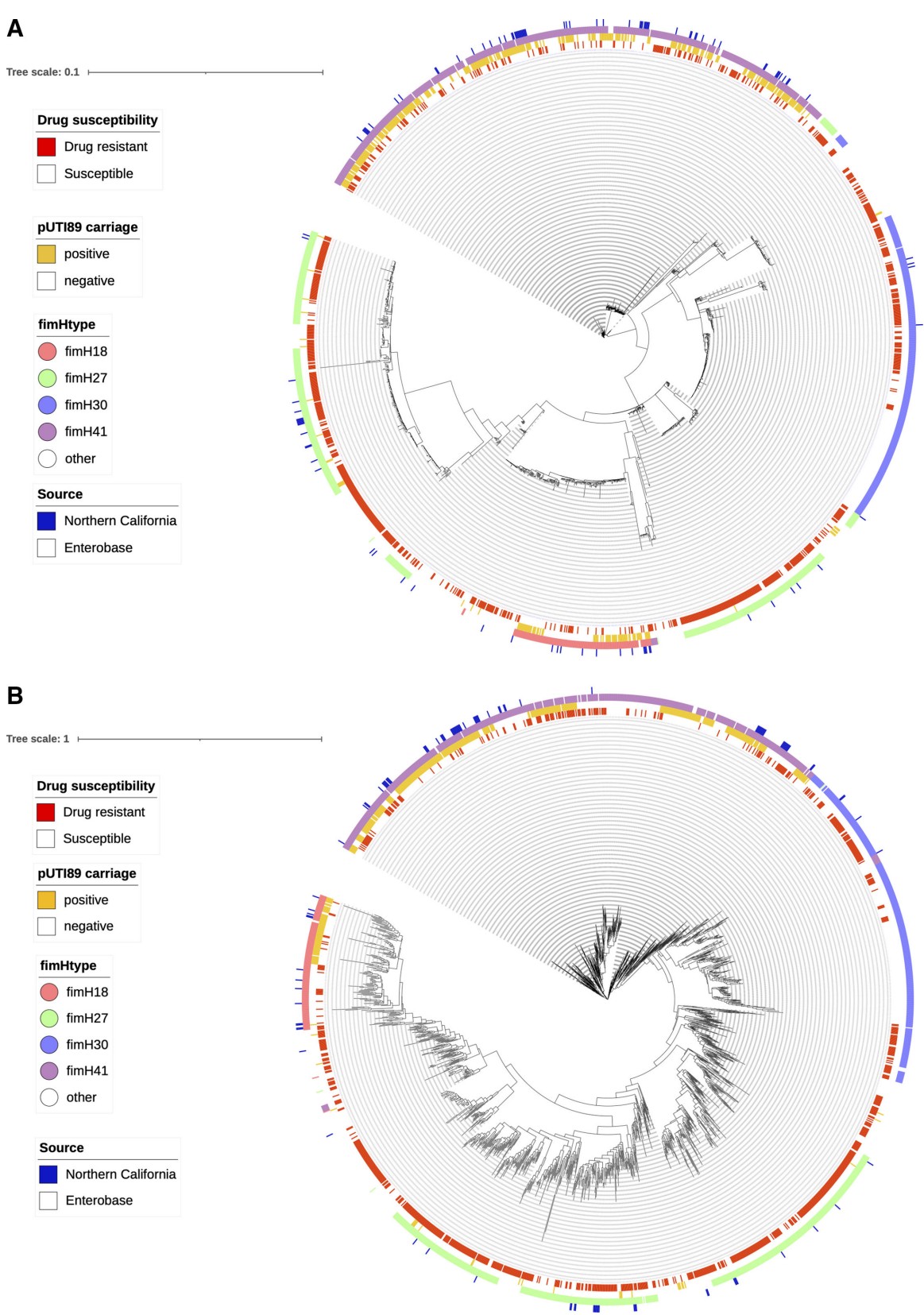

**FIG 1** Maximum-likelihood phylogeny of 1,749 ST95 isolates constructed with FastTree and visualized with iTOL (A) Phylogeny based on the core genome. (B) Phylogeny based on the presence and absence of accessory genes.

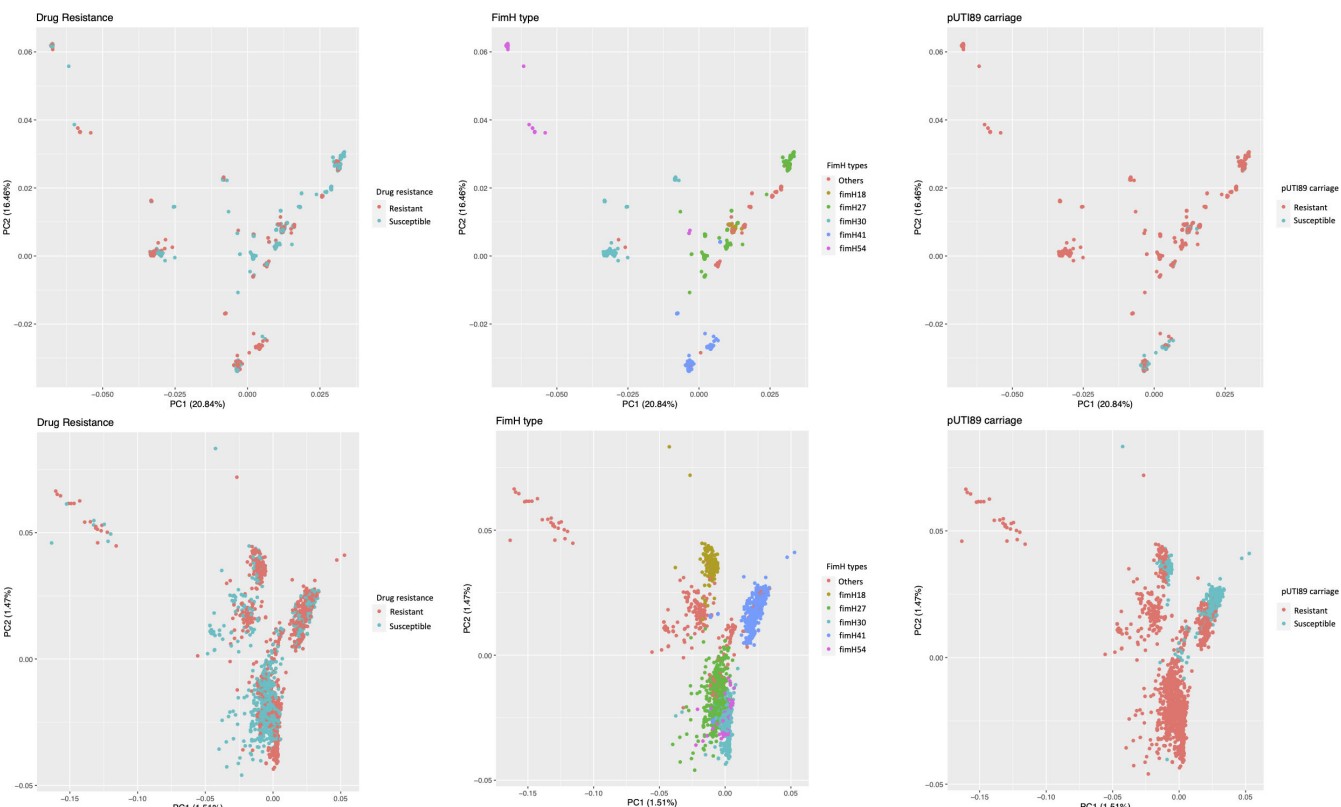

**FIG 2** Principal component analysis of 1,749 ST95 isolates based on (A) SNPs on the core genome and (B) presence and absence of the accessory genome.

## Genes associated with drug-susceptible ST95 strains

First, we conducted a univariate analysis with Fisher's exact test of 31,546 accessory genes on the entire collection of 1,749 ST95 isolates to identify genes uniquely associated with drug-susceptible strains. Due to the number of tests, Bonferroni-adjusted *P*-value was used to determine a statistical significance cut-off. We found that 884 accessory genes were significantly associated with drug susceptibility and 1,567 genes were significantly associated with drug resistance (Table S4). We also conducted a multivariate analysis adjusting for the carriage of pUTI89* and *fimH* types to control for confounding factors due to their previously reported significant association with drug susceptibility (21). We found that 154 accessory genes were significantly associated with drug susceptibility and 441 genes were significantly associated with drug resistance (Table S4).

We then conducted a genome-wide analysis with 261 UTI ST95 isolates and 292 BSI ST95 isolates (80 from Northern California and 473 from the EnteroBase ST95 collection) based on the metadata information. We limited the analysis to human BSI and UTI isolates (553 isolates in total) to preclude potential bias or errors that could result from including ST95 samples collected from unknown sources or sources other than human extraintestinal infections (e.g., environment, food animals, and human non-infection sources) indicated in the EnteroBase database. Genome-wide association analysis was done at the level of SNPs among core genes as determined with SNP sites (26), and at the level of gene content, as determined with Roary (27). A logistic multivariate regression model was conducted to identify genomic features significantly associated with drug susceptibility. In this model, the carriage of pUTI89* and *fimH* types was adjusted to control for confounders (21).

We found 44 accessory genes significantly associated with strains that lacked recognized drug resistance genes included in the ResFinder database (Table 2). We found 13 SNPs on core genes and 145 accessory genes associated with drug resistance (Table

**TABLE 2** Genes significantly associated with ST95 strains lacking drug resistance genes[a]

| Gene/SNP | Annotation | location | UTI (n = 261) | | BSI (n = 292) | | Entire ST95 (n = 1749) | | | ST131 |
| --- | --- | --- | --- | --- | --- | --- | --- | --- | --- | --- |
| | | | OR [95% CI] | P-value | OR [95% CI] | P-value | OR [95% CI] | P-value | Frequency (%) | |
| alkA | DNA-3-methyladenine glycosylase 2 | chromosome | 5.5 [1.7–20.4] | 0.007 | 2.5 [1–6.6] | 0.046 | 5.6 [4.2–7.5] | 0 | 1420 (80) | 5 |
| cia | colicin | plasmid | 4.8 [1.8–13.9] | 0.003 | 2.6 [1.1–6.2] | 0.023 | 2 [1.5–2.6] | 0 | 318 (18) | 0 |
| dinG_1 | 3'–5' exonuclease | plasmid | 5.3 [2–15.4] | 0.001 | 4.7 [1.9–12.3] | 0.001 | 2.3 [1.7–3] | 0 | 311 (18) | 1 |
| livK | leucine-specific binding protein precursor, high-affinity branched-chain amino acid ABC transporter substrate-binding protein LivK | chromosome | 3.8 [1.2–13.5] | 0.03 | 4 [1.5–12] | 0.008 | 0.9 [0.7–1.2] | 0.357 | 1546 (87) | 5 |
| manX_3 | PTS sugar transporter subunit IIA | chromosome | 3.8 [1.2–13.5] | 0.03 | 3.9 [1.5–11.8] | 0.009 | 0.9 [0.6–1.2] | 0.320 | 1547 (87) | 0 |
| mltC | membrane-bound lytic murein transglycosylase MltC | chromosome | 2.1 [1–4.4] | 0.042 | 2.1 [1–4.4] | 0.043 | NA [NA-36.1] | 0.481 | 1336 (75) | 1 |
| traD | type IV conjugative transfer system coupling protein TraD | plasmid | 4.1 [1.9–9.2] | 0 | 2.2 [1.2–4.3] | 0.014 | NA [NA-36.1] | 0.481 | 760 (43) | 1 |
| umuD_2 | translesion error-prone DNA polymerase V autoproteolytic subunit | pUTI89 | 11.2 [3.6–38.5] | 0 | 4.2 [1.6–10.9] | 0.003 | 5.3 [4–7.1] | 0 | 371 (21) | 1 |
| group_406 | conjugal transfer mating pair stabilization protein TraG | plasmid | 6.1 [2.5–16.1] | 0 | 4.5 [2.2–9.7] | 0 | 2.1 [1.6–2.7] | 0 | 379 (21) | 22 |
| group_1304 | hypothetical protein | pUTI89 | 43.5 [7.9–1000] | 0 | 9.7 [2.5–62.5] | 0.004 | 11.1 [6.7–19.4] | 0 | 196 (11) | 28 |
| group_2195 | DUF4113 domain-containing protein | plasmid | 12 [4–40] | 0 | 6.7 [1.9–24.4] | 0.003 | 7.6 [5.5–10.5] | 0 | 462 (26) | 0 |
| group_2212 | DNA repair protein UmuC | pUTI89 | 18.9 [5.5–90.9] | 0 | 5.8 [2–19.6] | 0.002 | 7 [4.8–10.3] | 0 | 264 (15) | 2 |
| group_2341 | incFII family plasmid replication initiator RepA | plasmid | 3.9 [1.6–10.2] | 0.004 | 3.4 [1.5–7.5] | 0.003 | 2 [1.6–2.5] | 0 | 377 (21) | 25 |
| group_2492 | conjugal transfer pilus acetylase TraX | plasmid | 4.9 [2–12.7] | 0.001 | 6.8 [3.3–15.2] | 0 | 1.9 [1.5–2.4] | 0 | 423 (24) | 25 |
| group_2899 | phage integrase family protein | pUTI89 | 21.7 [6.2–83.3] | 0 | 6.7 [1.9–24.4] | 0.003 | 8 [5.8–11.3] | 0 | 352 (20) | 28 |
| group_3175 | DUF945 domain-containing protein | pUTI89 | 6.6 [1.7–43.5] | 0.017 | 4.1 [1.1–19.6] | 0.048 | 5.1 [3.5–7.8] | 0 | 191 (11) | 23 |
| group_3361 | transglycosylase SLT domain-containing protein | pUTI89 | 6.6 [1.9–30.3] | 0.006 | 3.3 [1.1–11.2] | 0.044 | 5.2 [3.6–7.7] | 0 | 218 (12) | 23 |
| group_3450 | fertility inhibition protein FinO | plasmid | 3.4 [1.4–8.8] | 0.009 | 3 [1.4–6.4] | 0.005 | 2 [1.5–2.5] | 0 | 384 (22) | 25 |
| group_3636 | type IV conjugative transfer system protein TraL | pUTI89 | 6.2 [1.6–41.7] | 0.022 | 5.6 [1.3–38.5] | 0.036 | 7.6 [4.7–12.7] | 0 | 169 (10) | 22 |
| group_4084 | phage tail protein | chromosome | 6.3 [1.2–50] | 0.041 | 4.2 [1.4–13.3] | 0.012 | 1.7 [1.3–2.1] | 0 | 311 (18) | 1 |
| group_4555 | TrbJ conjugal transfer protein | plasmid | 6.4 [2.5–17.5] | 0 | 4.5 [2.1–9.7] | 0 | 2 [1.6–2.6] | 0 | 395 (22) | 12 |
| group_5037 | IS21 family transposase | pUTI89 | 23.3 [4.3–500] | 0.003 | 9.3 [2.3–62.5] | 0.005 | 13.2 [7.7–24.3] | 0 | 191 (11) | 5 |
| group_5504 | nuclease-like protein | plasmid | 4.8 [2–12.2] | 0.001 | 4.7 [2.4–9.6] | 0 | 1.8 [1.4–2.2] | 0 | 495 (28) | 9 |
| group_6014 | IncF conjugal transfer surface exclusion protein TraT | plasmid | 4 [1.7–9.4] | 0.001 | 4.5 [2.2–9.3] | 0 | 1.9 [1.5–2.4] | 0 | 446 (25) | 23 |
| group_6737 | phage minor tail protein G | chromosome | 2.6 [1.2–5.9] | 0.015 | 2.5 [1.2–5.3] | 0.018 | 1.3 [1.1–1.6] | 0.010 | 795 (45) | 0 |

*(Continued on next page)*

**TABLE 2** Genes significantly associated with ST95 strains lacking drug resistance genes[a] (*Continued*)

| Gene/SNP | Annotation | location | UTI (n = 261) | | BSI (n = 292) | | Entire ST95 (n = 1749) | | | ST131 |
|---|---|---|---|---|---|---|---|---|---|---|
| | | | OR [95% CI] | P-value | OR [95% CI] | P-value | OR [95% CI] | P-value | Frequency (%) | |
| group_6764 | hypothetical protein | plasmid | 4.8 [1.6–17.5] | 0.008 | 4 [1.3–15.6] | 0.027 | 9.8 [6–16.8] | 0 | 187 (11) | 28 |
| group_7168 | tail fiber domain-containing protein | chromosome | 6.3 [1.2–50] | 0.041 | 4.3 [1.4–15.2] | 0.016 | 1.8 [1.4–2.3] | 0 | 305 (17) | 1 |
| group_8375 | ISL3-like element ISEc53 family transposase | pUTI89, chromosome | 16.4 [4.1–111.1] | 0 | 8.9 [2.5–41.7] | 0.002 | 11.6 [7.3–19] | 0 | 238 (13) | 4 |
| group_8556 | phage portal protein | chromosome | 2.6 [1.2–5.9] | 0.015 | 2.5 [1.2–5.4] | 0.014 | 1.3 [1.1–1.6] | 0.010 | 603 (34) | 0 |
| group_9933 | DinI-like family protein | pUTI89, chromosome | 10.8 [3.3–38.5] | 0 | 4.8 [1.7–14.1] | 0.003 | 5.4 [4–7.2] | 0 | 373 (21) | 1 |
| group_9939 | malate dehydrogenase | plasmid | 3 [1.3–7.2] | 0.011 | 4.2 [2–9.3] | 0 | 1.5 [1.2–1.9] | 0 | 400 (23) | 0 |
| group_9942 | hypothetical protein | plasmid | 4.1 [1.8–10] | 0.001 | 3.8 [1.9–7.8] | 0 | 1.5 [1.2–1.8] | 0 | 459 (26) | 0 |
| group_9945 | hypothetical protein | plasmid | 4.1 [1.8–10.2] | 0.001 | 3.3 [1.5–7.5] | 0.003 | 1.3 [1–1.6] | 0.032 | 382 (22) | 0 |
| group_10528 | ATP-binding protein | pUTI89 | 19.6 [3.7–333.3] | 0.005 | 16.1 [3–333.3] | 0.009 | 16 [8.8–31.8] | 0 | 183 (10) | 1 |
| group_11645 | DNA breaking–rejoining protein | chromosome | 2.6 [1.2–5.9] | 0.015 | 2.5 [1.2–5.3] | 0.018 | 1.3 [1.1–1.6] | 0.009 | 610 (34) | 0 |
| group_12425 | PAS domain-containing protein | pUTI89 | 4.5 [1.4–17.2] | 0.016 | 9.2 [2.2–62.5] | 0.007 | 4.7 [3.3–6.8] | 0 | 226 (13) | 7 |
| group_13166 | PTS sugar transporter subunit IIB | chromosome | 3.8 [1.2–13.5] | 0.03 | 3.7 [1.5–10.5] | 0.009 | 0.9 [0.6–1.2] | 0.322 | 1545 (87) | 0 |
| group_13675 | ClbS/DfsB family four-helix bundle protein | pUTI89, chromosome | 10 [2.9–47.6] | 0.001 | 9.3 [2.7–43.5] | 0.001 | 11.5 [7.4–18.8] | 0 | 246 (14) | 4 |
| group_14237 | surface exclusion protein | plasmid | 3.1 [1.4–7] | 0.005 | 4.9 [2.5–10.1] | 0 | 1.7 [1.4–2.1] | 0 | 528 (30) | 0 |
| group_14238 | phage tail assembly protein T | chromosome | 2.6 [1.2–6] | 0.017 | 2.3 [1.1–5.2] | 0.033 | 1.4 [1.1–1.7] | 0.003 | 542 (31) | 0 |
| group_14249 | hypothetical protein | plasmid | 3.7 [1.6–8.8] | 0.002 | 4.5 [2.2–9.2] | 0 | 1.6 [1.3–2] | 0 | 485 (27) | 0 |
| group_15802 | replication regulatory protein RepA | plasmid | 3.2 [1.4–7.7] | 0.009 | 5.2 [2.6–10.8] | 0 | 1.5 [1.2–1.9] | 0 | 522 (29) | 4 |
| group_17896 | DUF2190 family protein | chromosome | 2.6 [1.2–5.9] | 0.015 | 2.5 [1.2–5.4] | 0.014 | 1.3 [1.1–1.6] | 0.009 | 597 (34) | 0 |
| group_20939 | peptidase M14 | plasmid | 3.2 [1.4–7.7] | 0.006 | 4.5 [2.2–9.2] | 0 | 1.6 [1.3–2] | 0 | 485 (27) | 0 |

[a]Annotation, location, and odds ratio with 95% CI are shown for UTI and BSI isolates. These ST95 genes that had a matching sequence in *Escherichia coli* O25b:H4-ST131 WGSs from the NCBI database are shown. Prevalence of these genes among 1,749 ST95 WGSs and their association with susceptible strains are also shown.

S5). The correlation coefficient matrix of these significantly associated genes is shown in Fig. 3.

Of the 44 genes associated with drug susceptibility strains, eight were annotated with Prokka as *alkA* (DNA-3-methyladenine glycosylase 2), *cia* (colicin), *dinG*_1 (3'–5' exonuclease), *livK* (ABC transporter substrate-binding protein LivK), *maX_3* (PTS sugar transporter subunit IIA), *mltC* (membrane-bound lytic murein transglycosylase MltC), *traD* (type IV conjugative transfer system coupling protein TraD), and *umuD_2* (trans-lesion error-prone DNA polymerase V autoproteolytic subunit). The other 36 genes were annotated by Prokka as encoding hypothetical proteins of unknown function. The sequences of these 36 hypothetical proteins were subjected to BLAST search. Of these, 30 were annotated in the NCBI database. Eight genes were annotated to encode the transfer system or transporter-related proteins, and five genes were annotated as phage genes. The annotation functions of the 44 genes are shown in Table 2.

## Distribution of genes associated with pan-susceptible ST95 strains

Of 44 accessory genes significantly associated with strains lacking antimicrobial resistance genes, 13 were found on pUTI89*, 19 were found on non-pUTI89* plasmids, and 15 were located in the chromosome (Table 2). Three genes were found both on the pUTI89* plasmid and chromosomes.

We also sought these 44 gene sequences in ExPEC strains belonging to ST131, which are largely resistant to one or more antimicrobial agents. Of these, 29 had a match ranging from 1 to 28 among 7,458 *Escherichia coli* O25b:H4-ST131 WGSs (Table S7). Thus, 15 genes associated with ST95 strains lacking drug resistance genes were not found in drug-resistant ST131 genomes. These genes included *cia*, *manX_3*, and 13 hypothetical proteins shown in Table 2. Of these 15 genes, 13 were significantly associated with

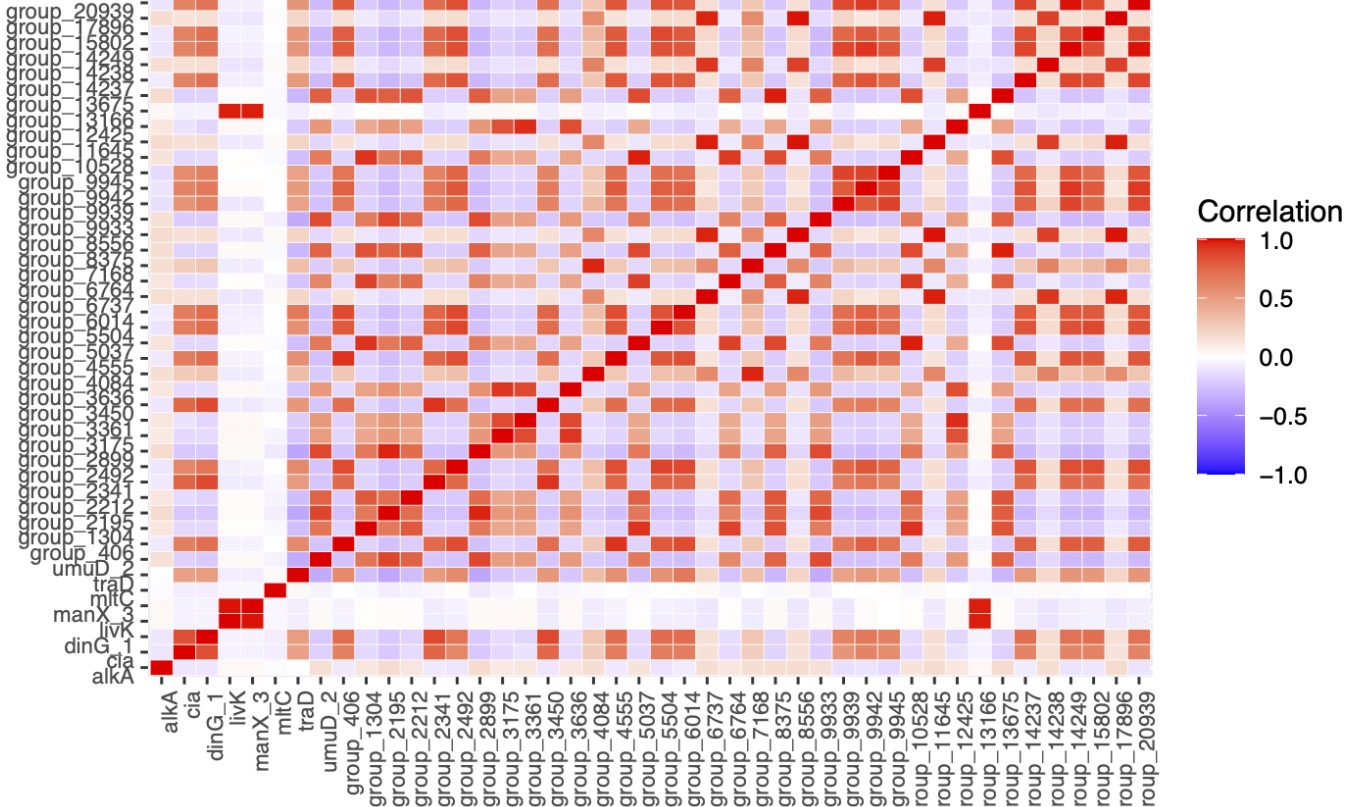

**FIG 3** The correlation coefficient matrix of ST95 genes significantly associated with ST95 strains lacking drug resistance genes. This illustrates which groups of genes associated with drug susceptibility are linked.

drug susceptibility among the entire 1,749 ST95 strains as well (Table 2). *manX_3* and group_13166 were not associated, which were annotated as PTS sugar transporter subunit IIA and PTS sugar transporter subunit IIB, respectively.

## Prophage analysis of ST95 Northern California isolates

Among the identified accessory genes significantly associated with drug susceptibility are included hypothetical phage genes (group_2899, group_4084, group_6737, group_8556, and group_14238). We analyzed eight ST95 Northern California isolates based on the presence and absence of these genes with PHASTER to identify potential phages from which these ST95 may have acquired the sequences. Phage gene group_4084 was not present among Northern California isolates. We found that seven of eight ST95 isolates contained intact Enterobacteria phage Sf101 (NCBI Reference Sequence: NC_027398) and *Shigella* phage SfII (NCBI Reference Sequence: NC_021857). Enterobacteria phage mEp460 (NCBI Reference Sequence: NC_019716) was found in five of eight ST95 isolates in this analysis. Two intact prophage sequences were found among two isolates, and four intact prophage sequences were found in single isolates (Table 3).

## DISCUSSION

We identified genomic features associated with strains lacking drug resistance genes of the ExPEC ST95 lineage isolated from BSI and UTI patients, as well as other sources. We analyzed 1,749 ST95 strains—1,669 ST95 from EnteroBase (downloaded on March 12, 2020) and 80 ST95 isolates from UTI or BSI patients in Northern California. We also conducted a genome-wide association study with 553 ST95 BSI and UTI human isolates. The sources of other isolates submitted to EnteroBase include soil, water, vegetables, animals other than humans, and human samples other than urine or blood (e.g., stool or wound). Our previous study of WGSs of 86 isolates of ST95 from BSI patients in the United States found that the carriage of UTI89-like plasmid (pUTI89*) or plasmid sequences and *fimH6* was significantly associated with drug-susceptible strains (21). We also reported that in Northern California, large proportions of ST95 strains remained susceptible to most of the drugs we tested compared to other major pandemic ExPEC lineages, despite its high prevalence among patients with community-acquired UTI (14). We hypothesized that ST95 strains can maintain competitive advantage under antibiotic stress without acquiring drug resistance genes among the human intestinal microbiota.

Of the 1,749 ST95 WGSs, 862 (49%) carried one or more drug resistance genes. Among these, 94 (36%) of 261 UTI ST95 isolates and 154 (53%) of 292 BSI ST95 isolates carried drug resistance genes. We observed a lower prevalence of drug-resistant ST95 isolates in Northern California than of those in the EnteroBase database (38% and 50%, respectively). This may be due to a submission bias to EnteroBase. For example, drug-resistant ST95 strains may have been more likely to be sequenced and deposited in EnteroBase. The Northern California isolates were population-based samples (14, 22).

Principal component analysis and phylogenetic analysis based on the SNPs of core genes and the presence and absence of accessory genes were conducted to identify potential confounding factors associated with clustering. In both analyses, carriage of pUTI89* and *fimH* types were strongly associated with clustering. This suggests that the carriage of pUTI89* and the *fimH* types explain the majority of genomic variation we extracted from the pan-genome analysis of the ST95 WGSs and that these two factors needed to be controlled in assessing the genetic features associated with the presence or absence of drug-resistant genes in conducting a pan-genome-wide association analysis. Clustering by *fim*H type is consistent with our previous findings (21) and also offers more evidence for *fimH* typing as a useful method for genotyping ExPEC (28).

We used multivariate logistic models to identify genomic features significantly associated with drug pan-susceptibility while controlling for *fimH* types and the carriage of pUTI89*. We limited the analysis to WGSs of those isolates from well-documented human BSI or UTI cases (553 isolates in total) to preclude potential bias that could result from including ST95 WGSs from unknown sources or non-human and non-human

**TABLE 3** Prophage analysis of 8 ST95 Northern California strains with different profiles of prophage sequences significantly associated with drug susceptibility[a]

| Drug resistance | Phage proteins identified as associated with drug susceptibility | Isolate | Number of prophage hits | Number of intact phage | PHAGE_Entero_ Sf101_NC_ 027398 | PHAGE_Shigel_ SfII_NC_ 021857 | PHAGE_Entero_ mEp460_NC_ 019716 | PHAGE_Entero_ BP_4795_NC_ 004813 | PHAGE_Klebsi_ 4LV2017_NC_ 047818 | Others |
|---|---|---|---|---|---|---|---|---|---|---|
| Susceptible | group_6737, group_8556, group_14238 | R007 | 9 | 4 | 1 | 1 | 1 | 1 | | |
| Susceptible | group_6737, group_8556, group_14238 | R757 | 8 | 2 | 1 | | 1 | | | |
| Resistant | group_6737, group_8556, group_14238 | R767 | 10 | 4 | 1 | 1 | 1 | | | PHAGE_Yersin_L_413C_NC_004745 |
| Susceptible | NA | R009 | 5 | 2 | 1 | 1 | | | | |
| Resistant | NA | R320 | 11 | 6 | 1 | 1 | | | 1 | PHAGE_Entero_P2_NC_001895 PHAGE_Salmon_SW9_NC_049459 PHAGE_Stx2_c_1717_NC_011357 |
| Susceptible | group_2899, group_6737, group_8556, group_14238 | R172 | 9 | 2 | 1 | 1 | 1 | | | |
| Susceptible | group_2899, group_6737, group_8556, group_14238 | R481 | 11 | 4 | 1 | 1 | 1 | | 1 | |
| Susceptible | group_2899 | R700 | 9 | 3 | 1 | 1 | | 1 | | |

[a]Drug resistance, presence of identified phage genes, isolate names, number of prophage hits, number of intact prophage sequences identified, and names and NCBI accession numbers of intact prophage sequences are shown.

infection sources. We initially found 44 accessory genes associated with drug susceptibility. We did not identify any SNPs in the core genes significantly associated with drug-susceptible strains. We also conducted a comprehensive analysis of these genes with other genotypes. Of these 44 genes, 15 were not found in any of 7,458 *E. coli* ST131 WGSs deposited in the NCBI nucleotide database. We compared with ST131 strains because ST131 strains are frequently multidrug-resistant and large number of WGSs in the NCBI database (15, 29). The other 29 genes showed one to 28 matches with the ST131 genome sequences, but these matches represented less than 1% of the ST131 WGSs, whereas they represented 10–87% of the ST95 WGSs (Table 2).

Thirteen of 15 genes remained associated with ST95 strains lacking drug resistance genes among the entire collection of WGSs we analyzed (Table 2). The two genes that lacked association among the entire collection of 1,749 WGSs were *manX_3* and group_13166, which were annotated as PTS sugar transporter subunit IIA and PTS sugar transporter subunit IIB, respectively.

Annotation of these 15 genes not found in any of the ST131 strains in the NCBI nucleotide database revealed that two genes were annotated as PTS sugar transporter subunit IIA and IIB, respectively. Previous reports revealed that PTS sugar transporters are involved in the penetration of polar antibiotics (30, 31). Three genes were annotated as phage genes. The other genes encoded colicin, malate dehydrogenase, DNA breaking–rejoining protein, surface exclusion protein, and peptidase M14. The five encoded hypothetical proteins had unknown functions.

Acquisition and maintenance of accessory genes by *E. coli* must provide a subpopulation of *E. coli* strains an adaptive advantage in an ecological niche. Prophage analysis revealed that the acquisition of phage proteins could play an important role in the acquisition of unique fitness genes (Table 3). ST95 strains may have evolved compensatory niche-adaptive mechanisms to remain viable in a variety of ecological niches (human and other animal intestine environment) despite remaining susceptible to antibiotics. One such mechanism is the role of amino acid transporters reported to contribute to metabolic adaptation of uropathogenic *E. coli* (UPEC) in the urinary tract (32).

A limitation of this study is that the drug resistance phenotype of 1,669 ST95 WGSs obtained from the EnteroBase database was predicted from the presence of drug resistance genes annotated in ResFinder and not by experimental phenotyping tests. Although genotypic resistance has been shown to have 100% sensitivity and specificity in predicting phenotypic resistance in *E. coli* by some studies (33, 34), we cannot rule out the possibility of misclassifying drug resistance in this study. Nevertheless, we found strong associations of multiple accessory genes with the absence of recognized drug resistance genes in ST95 strains.

Another limitation is the lack of consistent meta-data in the EnteroBase database. Although the EnteroBase database had data on 1,669 ST95 isolates, the majority of them lacked information on the source of ST95 strains, which led to the reduction of sample size for the genome-wide association study ($n = 553$). Also, the 80 ST95 isolates from Northern California were sequenced via a 300 bp paired-end run, not sufficient to yield a complete circle genome assembly. Still, our assembly data satisfied the quality test and were robust enough to assess the presence and absence of genes. Lastly, submission bias of ST95 to EnteroBase could have affected the prevalence of drug-resistant strains in our collection. The prevalence of drug-resistant ST95 strains in this collection may over-estimate the actual prevalence of drug-resistant ST95 in the human population.

In conclusion, we identified several genes associated with drug pan-susceptibility in ExPEC ST95 strains, which suggests that the ST95 lineage may have evolved to compensate for its susceptibility to antibiotics by acquiring unique fitness genes that enable a subpopulation of them to survive in multiple environmental niches without having to gain drug resistance. Experimental confirmation of these findings is needed to support this proposal, which is feasible with the relatively small number of unique genes in pan-susceptible ST95 strains we identified.

## MATERIALS AND METHODS

### Sample collection in Northern California

As previously described by Yamaji et al. (14) and Adams-Sapper et al. (22), we consecutively collected urine samples from patients with symptoms of UTI attending our university outpatient health service between September 2016 and May 2017 and all BSI isolates from inpatients admitted to San Francisco General Hospital (SFGH) between July 2007 and September 2010. All *E. coli* isolates were genotyped by multilocus sequence typing (MLST) based on the seven-gene scheme described at the PubMLST website (https://pubmlst.org/bigsdb?db//pubmlst_mlst_seqdef) (35). The seven-gene PCR products were sequenced by MiSeq System (Illumina) at the Functional Genomics Laboratory at University of California at Berkeley. All *E. coli* subtyped as ST95 were selected for WGS analysis. We analyzed 36 ST95 isolates from patients with UTI and 44 ST95 isolates from patients with BSI.

### Short read genome sequencing

Genomic DNA was extracted from all strains with the Qiagen blood and tissue DNeasy kit. Library preparation for the MiSeq platform following a standard protocol for Illumina-compatible libraries was conducted at Functional Genomics Laboratory in Berkeley. Final libraries were sequenced via a 300 bp paired-end run on a MiSeq instrument with V3 chemistry and standard Illumina analysis software at The California Institute for Quantitative Biosciences at UC Berkeley (QB3-Berkeley).

### Genome assembly

MiSeq reads were screened and trimmed based on the length and quality with BBDUK version 1.0 under the default setting (http://jgi.doe.gov/data-and-tools/bb-tools/). The trimming process also removed residual adapter sequences. Quality check of individual FASTQ files was conducted with FastQC (36), and all of the newly sequenced WGS data passed the quality check. *De novo* assembly of trimmed paired reads for the 80 libraries was performed with Unicycler version 0.4.8 under the setting "--min_fasta_length 500" to remove contigs less than 500 bp (37). The quality of *de novo* assembly was checked with QUAST version 5.0.2 under the default setting (38).

### Complementary dataset

We obtained 1,669 assembled whole genomes of ST95 and associated metadata from EnteroBase (24) accessed on March 12, 2020. Information on source type, source details, collection year, collection continent, and collection country of these whole genomes of ST95 is shown in Table S1.

### Bioinformatics analysis

Using the ResFinder database, we identified antimicrobial drug resistance (AMR) genes among the 1,749 assembled genomes of ST95. We defined drug-resistant strains as those having at least one recognized drug resistance gene and pansusceptible strains as those not having any recognized drug resistance gene. Plasmid incompatibility group and replicon types were identified with the PlasmidFinder database (https://bitbucket.org/genomicepidemiology/plasmidfinder_db/src/master/, accessed April 2020) (39). *FimH* types were identified with the FimTyper database (https://bitbucket.org/genomicepidemiology/fimtyper/src/master/, accessed April 2020) (40). All identifications were done with ABRicate version 1.0.1 (41) with a 95% identity threshold across the reference sequences.

Annotation was performed on all the assembled genomes with Prokka version 1.14.0 under the default setting (42). Genes annotated as encoding hypothetical proteins were further analyzed on BLAST (accessed April 2021) (43). A pangenome of the entire data

set was constructed with Roary version 3.13.0 (27) with a 95% identity cutoff. Here, genes present in 95% or more of the cohort isolates were defined as core and constituting the core portion of the pangenome, and genes present in less than 95% were defined as accessory and constituting the accessory portion of the pangenome. Accessory genes present in <15% of the cohort isolates were defined as cloud genes. A concatenated core CDS alignment ("core_gene_alignment.aln") was obtained from the Roary default output, and we extracted single-nucleotide polymorphism (SNP) information with snp-sites with the default option (26). Phylogenetic trees were constructed with FastTree (25) with the maximum likelihood method with the GTR model based on the SNP alignment generated with snp-sites and the presence and absence of accessory genes from the Roary default output ("accessory_binary_genes.fa"). Visualization was done with iToL version 6.1.2 (http://itol.embl.de).

## Statistical analysis

All statistical analyses were conducted with R version 3.6.3 (44). A *P*-value of <0.05 was considered statistically significant. We used the prcomp function for principal component analysis. The association of a specific accessory gene with drug resistance was first tested with Fisher's exact test without adjusting for any potential confounding factors with a cut-off at Bonferroni adjustment *P*-value and then tested with multivariate analysis with the glm function adjusted for the carriage of UTI89 like plasmid and *fimH* types. For the regression analysis, 553 ST95 isolates from BSI and UTI patients (473 from Enterobase and 80 from Northern California) were used to remove potential bias or errors that could occur from including ST95 samples collected from non-human or non-disease sources. Relationships between each set of significantly associated genetic features were assessed with correlation coefficients. All graphs were plotted with ggplot2 (45).

## Distribution of genes associated with pan-susceptibility

Once we identified ExPEC ST95 genes significantly associated with susceptibility to antimicrobial drugs, we analyzed the distribution and location of the gene in ST95 and other ExPEC STs by BLAST (accessed June 2021) (43). To identify whether the identified genes were located on pUTI89*, we sought for these genes in the pUTI89 complete sequence (GenBank accession number CP000244) on BLAST. We targeted ST131 because of their high frequency of drug resistance and the number of WGSs in the database (18, 46). We sought for these ST95 genes in (1) the ST131 chromosome, plasmid, or phages and (2) ST131 whole-genome sequences deposited in the NCBI database. NCBI has 127 complete whole-genome sequences of *Escherichia coli* O25b:H4-ST131 and 7331 plasmid or short genome sequences of *Escherichia coli* O25b:H4-ST131 deposited.

## Prophage analysis of ST95 Northern California isolates

Among the identified ExPEC ST95 genes associated with susceptibility to antimicrobial drugs were hypothetical phage proteins. We conducted a prophage analysis with PHASTER (47) to obtain the characteristics of the phage to identify a phage(s) from which ST95 may have acquired these genes. We selected *de novo* assemblies of eight Northern California ST95 isolates based on the profiles of these accessory genes and the drug susceptibility for PHASTER analysis.

## ACKNOWLEDGMENTS

We thank Holly Berryman Stern and the other staff members of the Tang Center of University of California, Berkeley (UC Berkeley)-affiliated health care service for their time and support of this project.

This work was supported by Centers for Disease Control and Prevention investments to combat antibiotic resistance under BAA 200–2016-91939. The contents of this work are solely the responsibility of the authors and do not necessarily represent the official

views of the Centers for Disease Control and Prevention. The funder of the study had no role in study design, data collection, data analysis, data interpretation, or writing of the report. The corresponding author had full access to all the data in the study and had final responsibility for the decision to submit for publication.

## AUTHOR AFFILIATIONS

[1]University of California Berkeley, Berkeley, California, USA
[2]World Health Organization, Geneva, Switzerland
[3]Resilient Biotics, San Francisco, California, USA

## AUTHOR ORCIDs

Yuan Hu Allegretti ⓘ http://orcid.org/0000-0002-3031-9341

## FUNDING

| Funder | Grant(s) | Author(s) |
|---|---|---|
| HHS | Centers for Disease Control and Prevention (CDC) | BAA 200-2016-91939 | Lee W. Riley |

## AUTHOR CONTRIBUTIONS

Yuan Hu Allegretti, Conceptualization, Data curation, Formal analysis, Investigation, Methodology, Project administration, Visualization, Writing – original draft, Writing – review and editing | Sheila Adams-Sapper, Conceptualization, Data curation, Investigation, Supervision, Writing – review and editing | Lee W. Riley, Conceptualization, Funding acquisition, Project administration, Resources, Supervision, Writing – review and editing.

## DATA AVAILABILITY

The whole-genome shotgun sequence results described here have been deposited in DDBJ/ENA/GenBank under the accession numbers shown in Table S6. Genome sequences have been deposited in NCBI BioProject under accession number PRJNA763994.

## ADDITIONAL FILES

The following material is available online.

### Supplemental Material

**Supplemental legends (Spectrum04189-22-s0001.docx).** Legends for Tables S1 to S7.
**Table S1 (Spectrum04189-22-s0002.csv).** Information on source type, source details, collection year, collection continent, and collection country of whole genomes of ST95 downloaded from EnteroBase.
**Table S2 (Spectrum04189-22-s0003.csv).** Drug resistance genes identified from the whole collection of 1749 ST95 from ResFinder.
**Table S3 (Spectrum04189-22-s0004.csv).** Frequency of drug resistance among ST95 isolates registered in the EnteroBase database between 1947 and 2019 and ST95 isolates from Northern California between 2013 and 2017, by year, stratified by isolation information (BSI or UTI).
**Table S4 (Spectrum04189-22-s0005.csv).** Univariate and multivariate analysis of the pangenome of 1,749 ST95 whole-genome sequences.
**Table S5 (Spectrum04189-22-s0006.csv).** Information of 13 SNPs on core genes and 145 accessory genes associated with drug resistance.
**Table S6 (Spectrum04189-22-s0007.csv).** Accession numbers of submitted ST95 whole-genome sequences in DDBJ/ENA/GenBank.

**Table S7 (Spectrum04189-22-s0008.csv).** NCBI accession numbers and the numbers of matching regions of ST131 whole-genome sequence that carry one or more genes of 44 ST95 genes found to be associated with ST95 strains lacking drug resistance genes.

## Open Peer Review

**PEER REVIEW HISTORY (review-history.pdf).** An accounting of the reviewer comments and feedback.

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
