## [Reviewer comments · Microbiology Spectrum]

Microbiology Spectrum

Genetic features of antimicrobial drug-susceptible extraintestinal pathogenic *Escherichia coli* pandemic sequence type 95

Yuan Allegretti, Reina Yamaji, Sheila Adams-Sapper, and Lee Riley

Corresponding Author(s): Yuan Allegretti, University of California Berkeley

Review Timeline:

Submission Date:	October 16, 2022
Editorial Decision:	March 15, 2023
Revision Received:	May 12, 2023
Editorial Decision:	July 22, 2023
Revision Received:	July 25, 2023
Accepted:	November 13, 2023

Editor: Neelam Taneja

Reviewer(s): Disclosure of reviewer identity is with reference to reviewer comments included in decision letter(s). The following individuals involved in review of your submission have agreed to reveal their identity: Edwin Barrios-Villa (Reviewer #1); Maria Tomas (Reviewer #2); Alan J Wolfe (Reviewer #3)

Transaction Report:

DOI: <https://doi.org/10.1128/spectrum.04189-22>

March 15, 2023

Dr. Yuan Hu Allegretti
University of California Berkeley
Berkeley

Re: Spectrum04189-22 (Genetic features of antimicrobial drug-susceptible extraintestinal pathogenic *Escherichia coli* pandemic sequence type 95)

Dear Dr. Yuan Hu Allegretti:

Link Not Available

Sincerely,

Neelam Taneja

Journals Department
Reviewer comments:

Reviewer #1 (Comments for the Author):

The authors presented a very well-written paper entitled "Genetic features of antimicrobial drug-susceptible extraintestinal pathogenic *Escherichia coli* pandemic sequence type 95" They compared 80 genomes of ST95 strains from BSI, UTI against 1669 deposited in a public database.

The work was well conducted, and the conclusions give an important rise to the field, my observations/suggestions are minimal and are as follows:

L106, please arase the blank space on the sentence "accessory genes ." to "accessory genes."

L219, please clarify why you are comparing against ST131. Logical and previous knowledge suggests a comprehensive comparison but could be fine to reference the Material and Methods section (4.7) where is widely justified.

All the sequenced genomes and plasmids were circled? or the assemblies are just scaffolds? when you use Illumina and trimming of the drafts it is possible to obtain short reads that used to be eliminated (as well as you did). These missing sequences made it impossible to close a genome with just one sequencing round. Please clarify and adjust the conclusions based on this asseveration.

Fig. 1A the "Data Source" color is difficult to see in the figure please consider changing

Fig 1B the "Data source" color is better but seems to be the same of Fig 1A, with the same observation.

Table 1 and table 2. Resistance genes names must be italicized

Reviewer #2 (Comments for the Author):

Review control no. Spectrum04189-22

The work titled "Genetic features of antimicrobial drug-susceptible extraintestinal pathogenic Escherichia coli pandemic sequence type 95" show the genomic analysis of high number of the clinical strains of the Escherichia coli ExPEC ST95 which have a susceptible background. The authors found a presence of the pUTI89 plasmids as well as fimH types to control significant association with drug-susceptibility. It would be necessary some improvements in the manuscript to publish in the Microbiology Spectrum:

1. To analyze all proteins (ORFs) from the pUTI89 plasmids and development a Figure comparison the genomic features from these important plasmids. To work in the annotation of the genes which constitute the plasmids. To make a Figure of the genomic comparison.
2. The Figures have low quality. Please, the authors must improve it.
3. There are phage proteins in the genomic analysis from Escherichia coli ExPEC ST95 core genome. Add the prophage study by bioinformatic tools: PHASTER, Prophinder, Prophage hunter,others. And to confirm the phage complete by TEM microscopy. Also, to make a Table or Figure with the proteins from prophages.
4. Is there database from Genbank of the genomes? Please, the authors must to make it (Bioproject) with the genomes or plasmids and prophages.

In conclusion, the study needs more bioinformatic and laboratory studies to improve the presentation of the results as well as the discussion.

Staff Comments:

Preparing Revision Guidelines

Please return the manuscript within 60 days; if you cannot complete the modification within this time period, please contact me. If you do not wish to modify the manuscript and prefer to submit it to another journal, please notify me of your decision immediately so that the manuscript may be formally withdrawn from consideration by Microbiology Spectrum.

Review control no. Spectrum04189-22

The work titled **“Genetic features of antimicrobial drug-susceptible extraintestinal pathogenic Escherichia coli pandemic sequence type 95”** show the genomic analysis of high number of the clinical strains of the Escherichia coli ExPEC ST95 which have a susceptible background. The authors found a presence of the pUTI89 plasmids as well as fimH types to control significant association with drug-susceptibility. It would be necessary some improvements in the manuscript to publish in the Microbiology Spectrum:

- 1.** To analyze all proteins (ORFs) from the pUTI89 plasmids and development a Figure comparison the genomic features from these important plasmids. To work in the annotation of the genes which constitute the plasmids. To make a Figure of the genomic comparison.

- 2.** The Figures have low quality. Please, the authors must improve it.

- 3.** There are phage proteins in the genomic analysis from Escherichia coli ExPEC ST95 core genome. Add the prophage study by bioinformatic tools: PHASTER, Prophinder, Prophage hunter,others. And to confirm the phage complete by TEM microscopy. Also, to make a Table or Figure with the proteins from prophages.

- 4.** Is there database from Genbank of the genomes? Please, the authors must to make it (Bioproject) with the genomes or plasmids and prophages.

In conclusion, the study needs more bioinformatic and laboratory studies to improve the presentation of the results as well as the discussion.

Dear Editor,

The revision of our manuscript “Genetic features of antimicrobial drug-susceptible extraintestinal pathogenic *Escherichia coli* pandemic sequence type 95” has been prepared, with a point by point response to reviewers below (reviewer comments are in bold). We thank our reviewers for their valuable advice and recommendations. We have addressed the suggestions and questions raised by the reviewers and highlighted the changes made in the revision. The changes in our manuscript are highlighted in yellow.

Sincerely,
Yuan Hu Allegretti (on behalf of all authors)

Reviewer Comments: Reviewer 1

The authors presented a very well-written paper entitled "Genetic features of antimicrobial drug-susceptible extraintestinal pathogenic *Escherichia coli* pandemic sequence type 95" They compared 80 genomes of ST95 strains from BSI, UTI against 1669 deposited in a public database.

The work was well conducted, and the conclusions give an important rise to the field, my observations/suggestions are minimal and are as follows:

L106, please erase the blank space on the sentence "accessory genes ." to "accessory genes."

The blank space was erased.

L219, please clarify why you are comparing against ST131. Logical and previous knowledge suggests a comprehensive comparison but could be fine to reference the Material and Methods section (4.7) where is widely justified.

Following your suggestion, we modified the discussion as follows:

“We also conducted a comprehensive analysis of these genes with other genotypes. Of these 44 genes, 15 were not found in any of 7458 *E. coli* ST131 WGSs deposited in the NCBI nucleotide database. We compared with ST131 strains because ST131 are frequently multidrug resistant and large number of WGSs in NCBI database.”

All the sequenced genomes and plasmids were circled? or the assemblies are just scaffolds? when you use Illumina and trimming of the drafts it is possible to obtain short reads that used to be eliminated (as well as you did). These missing sequences made it impossible to close a genome with just one sequencing round. Please clarify and adjust the conclusions based on this asseveration.

The assemblies were scaffolds, and we could not obtain circled assemblies. We clarified this and added the following text to the discussion section:

“Also, the 80 ST95 isolates from Northern California were sequenced via a 300-bp paired-end run, not sufficient to yield a complete circled genome assembly. Still, our assembly data satisfied the quality test and was robust enough to assess the presence and absence of genes.”

Fig. 1A the "Data Source" color is difficult to see in the figure please consider changing Fig 1B the "Data source" color is better but seems to be the same of Fig 1A, with the same observation.

We changed the color from pink to dark blue.

Table 1 and table 2. Resistance genes names must be italicized

We italicized the resistance genes names in Table 1 and the genes names in Table 2.

Reviewer Comments: Reviewer 2

The work titled "Genetic features of antimicrobial drug-susceptible extraintestinal pathogenic *Escherichia coli* pandemic sequence type 95" show the genomic analysis of high number of the clinical strains of the *Escherichia coli* ExPEC ST95 which have a susceptible background. The authors found a presence of the pUTI89 plasmids as well as fimH types to control significant association with drug-susceptibility. It would be necessary some improvements in the manuscript to publish in the *Microbiology Spectrum*:

1. To analyze all proteins (ORFs) from the pUTI89 plasmids and development a Figure comparison the genomic features from these important plasmids. To work in the annotation of the genes which constitute the plasmids. To make a Figure of the genomic comparison.

We appreciate your valuable suggestions. We would like to emphasize that our major findings in this research are identification of ST95 accessory genes significantly associated with drug susceptibility, which may help shed new light on the compensatory fitness advantage of ST95 under antimicrobial drug stress without acquiring drug-resistance genes. Presence of the UTI89 like plasmid (pUTI89*) and fimH types were found to be significantly associated with the drug susceptibility of ST95 isolates in our previous research (Stephens et al., Genomic Analysis of Factors Associated with Low Prevalence of Antibiotic Resistance in Extraintestinal Pathogenic *Escherichia coli* Sequence Type 95 Strains, *mSphere* 2:e00390-16. doi: 10.1128/mSphere.00390-16). Here, we controlled the carriage of pUTI89* and fimH types to avoid confounding in our pangenome-wide association study.

While we understand the importance of obtaining the complete sequence of pUTI89*, we submitted our isolates DNA to 300-bp paired-end run to obtain whole genome sequences, which is not sufficient to obtain complete circled plasmid sequences. We identified the carriage of pUTI89* by observing the replicon sequence unique to pUTI89* with the PlasmidFinder. We assessed whether the accessory gene was located on pUTI89* or not by searching on BLAST with complete pUTI89 reference sequence (Genbank accession number CP000244). Since this was not very clear from our initial manuscript, we modified the Material and Methods section 4.7 to improve clarity.

We annotated our entire collection of ST95 whole genome sequences (Material and Methods 4.5), which included all the proteins from the pUTI89 plasmids as well. We used the annotation results to conduct a pangenome analysis and then compared all of the accessory genes (31,546 genes). This included all the accessory genes located on pUTI89 plasmid in addition to the ones located on chromosome and other small plasmids. The result of the genomic comparison of these accessory genes between resistant isolates and susceptible isolates are available on Supplementary Table 4 and Supplementary Table 5.

2. The Figures have low quality. Please, the authors must improve it.

Following the reviewers' advice, we changed the color of Figure 1A and Figure 1B to make it easier for readers to see. We also changed the format of Figure 2 and Figure 3 for better quality. We modified the legends of Figure 2 (PCA analysis). Figure 3 (correlation matrix) was modified to include 44 accessory genes significantly associated with the drug susceptibility, not all the genes associated with either drug resistance or drug susceptibility.

3. There are phage proteins in the genomic analysis from Escherichia coli ExPEC ST95 core genome. Add the prophage study by bioinformatic tools: PHASTER, Prophinder, Prophage hunter,others. And to confirm the phage complete by TEM microscopy. Also, to make a Table or Figure with the proteins from prophages.

We conducted genomic analysis from ExPEC ST95 accessory genome, which are shared among less than 95% of the isolates, not from ST95 core genome (shared among more than 95% of the isolate). Still, we understand the importance of adding the prophage study to confirm phage sequences, and we therefore selected isolates with different phage protein profiles and drug resistance phenotypes, conducting prophage analysis with PHASTER. The results are shown in Table 3, and we adjusted the methods (Section 4.8), results (Section 2.6), and discussion sections based on this new data.

While we understand that experiments with TEM microscopy can provide additional confirmation of phage particles, multiple recent prophage analysis studies did not conduct TEM microscopy to identify phage(s) in addition to the prophage analysis using whole genome sequence data with computational bioinformatics tools. Some examples of this from the recent literatures are listed below. Given the widely recognized reliability of

PHASTER, prophage analysis with sequence data should be sufficient to predict the phage from which the prophage sequences came from.

We would also like to emphasize that the highlight of this work was the identification of accessory genes associated with drug susceptibility. Morphology or particulate interactions of these genes could be further investigated as a continuation of this work in the future.

1. Sharma, V., Hünnefeld, M., Luthe, T. et al. Systematic analysis of prophage elements in actinobacterial genomes reveals a remarkable phylogenetic diversity. *Sci Rep* 13, 4410 (2023). doi: 10.1038/s41598-023-30829-z
2. Dominguez-Mirazo M, Jin R, Weitz JS. Functional and Comparative Genomic Analysis of Integrated Prophage-Like Sequences in "Candidatus *Liberibacter asiaticus*". *mSphere*. 2019 Nov 13;4(6):e00409-19. doi: 10.1128/mSphere.00409-19.
3. Gao R, Duceppe MO, Chattaway MA, Goodridge L, Ogunremi D. Application of prophage sequence analysis to investigate a disease outbreak involving *Salmonella Adjame*, a rare serovar and implications for the population structure. *Front Microbiol*. 2023 Mar 3;14:1086198. doi: 10.3389/fmicb.2023.1086198.

4. Is there database from Genbank of the genomes? Please, the authors must to make it (Bioproject) with the genomes or plasmids and prophages.

We have submitted all of our newly sequenced genomes to Genbank, as described in Materials and Methods section 4.9 (section 4.8 in the original manuscript):

“The whole-genome shotgun sequence results described here have been deposited in DDBJ/ENA/GenBank under the accession numbers shown in Supplementary Table 6. Genome sequences have been deposited in NCBI BioProject under accession number PRJNA763994.”

Also, the metadata of all other whole genome sequences used in this project and obtained from Enterobase is shown in Supplementary Table 1.

July 22, 2023

Dr. Yuan Hu Allegretti
University of California Berkeley
Berkeley

Re: Spectrum04189-22R1 (Genetic features of antimicrobial drug-susceptible extraintestinal pathogenic *Escherichia coli* pandemic sequence type 95)

Dear Dr. Yuan Hu Allegretti:

Thank you for submitting your manuscript to Microbiology Spectrum. As you will see your paper is very close to acceptance. Please modify the manuscript along the lines I have recommended. As these revisions are quite minor, I expect that you should be able to turn in the revised paper in less than 30 days, if not sooner. If your manuscript was reviewed, you will find the reviewers' comments below.

When submitting the revised version of your paper, please provide (1) point-by-point responses to the issues raised by the reviewers as file type "Response to Reviewers," not in your cover letter, and (2) a PDF file that indicates the changes from the original submission (by highlighting or underlining the changes) as file type "Marked Up Manuscript - For Review Only". Please use this link to submit your revised manuscript. Detailed instructions on submitting your revised paper are below.

Link Not Available

Sincerely,

Neelam Taneja

Reviewer comments:

Reviewer #1 (Comments for the Author):

All the observations were properly addressed

Reviewer #3 (Comments for the Author):

I have just a couple of suggested edits and a question.

Line 61: "...more than 50%..." is grammatically correct.

Line 83: Where is supplementary table 1 listed in the text? Did I miss it?

Line 148: Should this be "confounders" rather than "confounding"?

Preparing Revision Guidelines

Please return the manuscript within 60 days; if you cannot complete the modification within this time period, please contact me. If you do not wish to modify the manuscript and prefer to submit it to another journal, please notify me of your decision immediately so that the manuscript may be formally withdrawn from consideration by Microbiology Spectrum.

Dear Editor,

The revision of our manuscript “Genetic features of antimicrobial drug-susceptible extraintestinal pathogenic *Escherichia coli* pandemic sequence type 95” has been prepared, with a point by point response to reviewers below (reviewer comments are in bold). We thank our reviewers for their valuable advice and recommendations. We have addressed the suggestions and questions raised by the reviewers and highlighted the changes made in the revision. The changes in our manuscript are highlighted in yellow.

Sincerely,
Yuan Hu Allegretti (on behalf of all authors)

Reviewer Comments: Reviewer 1

All the observations were properly addressed

Thank you so much.

Reviewer Comments: Reviewer 3

I have just a couple of suggested edits and a question.

Line 61: "...more than 50%..." is grammatically correct.

Line 148: Should this be "confounders" rather than "confounding"?

We fixed the sentences accordingly.

Line 83: Where is supplementary table 1 listed in the text? Did I miss it?

We checked the Manuscript Items listed in the portal and confirmed all the supplementary tables were present. We believe Supplementary tables were not included in the merged pdf for reviewers.

Re: Spectrum04189-22R2 (Genetic features of antimicrobial drug-susceptible extraintestinal pathogenic *Escherichia coli* pandemic sequence type 95)

Dear Dr. Yuan Hu Allegetti:

Your manuscript has been accepted, and I am forwarding it to the ASM production staff for publication. Your paper will first be checked to make sure all elements meet the technical requirements. ASM staff will contact you if anything needs to be revised before copyediting and production can begin. Otherwise, you will be notified when your proofs are ready to be viewed.

Sincerely,
Neelam Taneja
Editor
Microbiology Spectrum

Reviewer #1 (Comments for the Author):

The paper gives important data to the field of the research of Extraintestinal *Escherichia coli*. The manuscript is well-written with scientific sound and the conclusions are supported by the results.